# A computationally efficient depression-filling algorithm for digital elevation models applied to proglacial lake drainage

Constantijn J. Berends[1], Roderik S. W. van de Wal[1]

[1]Institute for Marine and Atmospheric research Utrecht, Utrecht University, Utrecht, 3584 CC, The Netherlands

5  *Correspondence to*: Constantijn J. Berends (c.j.berends@uu.nl)

**Abstract.** Many processes govern the deglaciation of ice sheets. One of the processes that is usually ignored is the calving of ice in lakes that temporarily surround the ice sheet. In order to capture this process a flood-fill algorithm is needed. Here we present and evaluate several optimizations to a standard flood-fill algorithm in terms of computational efficiency. As an example, we determine the land/ocean-mask for a 1 km resolution digital elevation model (DEM) of North America and 10  Greenland, a geographical area of roughly 7000 by 5000 km (roughly 35 million elements), about half of which is covered by ocean. Determining the land/ocean-mask with our improved flood-fill algorithm reduces computation time by 90 % relative to using a standard stack-based flood-fill algorithm. This implies that it is now feasible to include the calving of ice in lakes as a dynamical process inside an ice sheet model. We demonstrate this by using bedrock elevation, ice thickness and geoid perturbation fields from the output of a coupled ice-sheet - sea-level equation model at 30,000 years before present and 15  determine the extent of Lake Agassiz, using both the standard and improved versions of the flood-fill algorithm. We show that several optimizations to the flood-fill algorithm used for filling a depression up to a water level, that is not defined at forehand, decrease the computation time by up to 99 %. The resulting reduction in computation time allows determination of the extent and volume of depressions in a DEM over large geographical grids or repeatedly over long periods of time, where computation time might otherwise be a limiting factor. The algorithm can be used for all glaciological and hydrological 20  models, which need to trace the evolution over time of lakes or drainage basins in general.

## 1 Introduction

Changes in lake extent over time play an important role in hydrology (Renssen and Knoop, 2000), palaeo climatology (Krinner et al., 2004, Goelzer et al., 2012) and glaciology (Marshall et al., 1999, Tarasov and Peltier, 2006). For this reason it is important to track such changes in water routing and drainage inside dynamical models, particularly if major changes in 25  the boundary conditions (topography, ice extent, ocean volume) take place over time. Several studies have underlined the importance of accounting for depressions when creating such drainage maps (Zhu et al., 2006, Arnold, 2010).

An important example is the routing of melt-water and the drainage of proglacial lakes that formed near the margins of the Laurentide ice-sheet. The retreat of the Laurentide ice-sheet during the last deglaciation and the corresponding release of large fluxes of fresh water into the Arctic Sea and the northern Atlantic Ocean have been linked to climatic events through

disrupting the Atlantic Meridional Overturning Circulation (Barber et al., 1999, Clark et al., 2001, Li et al., 2012, Tarasov and Peltier, 2005, Teller et al., 2002). Melt-water from the remnants of Northern Hemisphere ice-sheets has also shown to have influenced the climate system during the previous interglacial (Stone et al., 2016).

Many studies have focused on reconstructing the direction and magnitude of the freshwater flux from geological data

(Broecker et al., 1989, Hillaire-Marcel et al., 2007, LaJeunesse and St.-Onge, 2008, Törnqvist et al., 2004). More recently these freshwater fluxes have been estimated by stand-alone ice-sheet models (Goelzer et al., 2012, Marshall et al., 1999, Tarasov and Peltier, 2006).

Due to the influence of the local topography, part of the melt-water from the retreating ice-sheet was not directly released into the ocean system but temporarily stored in proglacial lakes, the largest of which is Lake Agassiz, along the southern

margin of the ice-sheet. Lake Agassiz is estimated to have covered $7.1 \times 10^5$ km$^2$ and contained approximately $1.4 \times 10^{14}$ m$^3$ of water or 0.4 m global mean sea-level equivalent around 8.4 kyr BP, immediately prior to its catastrophic drainage (Kendall et al., 2008). The size of the flood is however poorly constrained and higher numbers have also been published (e.g. Hijma and Cohen, 2010). It is therefore important to accurately model the extent and volume of the lake over time, since the presence of the lake affects both the timing and location of the melt-water release into the ocean system, as well as the local

climate (Krinner et al. 2004, Tarasov and Peltier, 2006).

In order to model the extent and volume of the lake and the drainage water flux and drainage location over time we first need to define a land/ocean-mask. This mask delineates areas below sea level, which are connected to the open ocean from the land points, which may change over time. Large changes occur for example, where the ice-sheet at times covers most of the Canadian Archipelago or block the Hudson Strait. Changes in this mask change the location where outflow from the lake

reaches the sea over time. Besides determining the land/ocean mask we need to fill the depressions in the surface topography to determine the extent and volume of the lake(s). Both these problems - determining a land/ocean-mask and determining the volume of a lake filling a depression - are solved using so-called "flood-fill algorithms". Given an array of elements that individually may or may not satisfy a certain threshold condition, a flood-fill algorithm determines which elements of the set have a neighbour-to-neighbour connection to a given "seed element" that passes only through elements that satisfy the

threshold condition. The land/ocean-mask, for example, consists of the set of elements that have such a connection to the open ocean, where the threshold condition is met when the surface elevation of an element lies below sea level.

When applying commonly used flood-fill algorithms (Arnold, 2010, Doll and Lehner, 2002, Tarboton et al., 1991, Zhu et al., 2006) to a problem involving such large geographical grids and long time-scales, computation time can become a limiting factor, particularly when the geometry is changing over time and the procedure has to be repeated over many time steps.

In this study, we describe and evaluate several improvements to a standard algorithm for filling depressions in a DEM in order to improve the computational efficiency. The improved algorithm is applied to a 1 km resolution DEM, including an ice thickness distribution of North America 30,000 years ago generated by an ice sheet model in such a way as to create boundary conditions allowing for the formation of a very large lake. We determine both the land/ocean-mask and the size and extent of the proglacial lake for this glacial configuration and compare the required computation time with the default

flood-fill algorithm as presented by Zhu et al. (2006) and with the drainage pointer approach presented by Tarasov and Peltier (2006).

**2 Methodology**

**2.1 Default algorithm**

5 The problem of filling a depression up to a pre-defined level can be envisioned as filling a hole in a true/false-mask (whether or not the local topography is below the a priori chosen water level of the lake or ocean. There are several algorithms for solving this problem, generally known as "flood-fill" algorithms (Zhu et al., 2006). They are commonly known for their use in the "bucket" tool of several paint programs. A thorough description of the stack-based flood-fill algorithm is given by Zhu et al. (2006). The default algorithm as used by Zhu et al. (2006) starts with a "seed", defined as a designated element from

10 where neighbouring elements are flooded. In many hydrological applications, the seed will be a local minimum in a DEM. Beside the seed element, the algorithm builds a "stack", which is an array listing the indices of the map elements. During each iteration of the algorithm, all stack elements are checked. If the elevation of a stack element is below the water level, the element is filled with water and it is removed from the stack and consequently all its neighbours are added to the stack, thus expanding the filled area outwards. If the element does not lie below the water level it is removed from the stack

15 without any further action. The iterations are continued until the stack is empty and the elements below the water level are identified.

Once the horizontal extent of the lake has thus been determined, its volume is calculated by integrating the water depth over the calculated area.

A pseudo-code example of this algorithm is illustrated below. A step-by-step illustration of one iteration of the algorithm

20 being applied to a simple true/false mask is shown in Figure 1.

```
stack = seed

WHILE (stack is not empty)
 FOR (all elements in stack)
  IF (stack element lies below water level)
   remove element from stack
   FOR (8 neighbours of stack element)
    IF (neighbour is not filled and is not in stack)
     add neighbour to stack
    END IF
   END FOR
  END IF
 END FOR
END WHILE
```

An example of a Matlab implementation of this algorithm is provided in the supplementary material, being script "fill_1km.m".

The number of operations required for filling a hole with this algorithm is approximately proportional to the number of elements of the depression. This means that application to a very large area with a high resolution results in a long computation time, since doubling the resolution in the horizontal plane will quadruple the number of operations.

Determining the land/ocean-mask of a DEM using this algorithm is straightforward, since the water level (the elevation of the water surface with respect to the Earth's centre) that determines the fill criterion for all elements is, by definition, at sea level. For a lake where the water inflow is not balanced by evaporation, the water level depends on the topography surrounding the depression. The water level is then equal to the elevation of the "spill-over point" - the origin of the river that transports spill-over from the lake to the open ocean. The only way to determine the location of this point, and therefore the water level of the lake, is to start at the chosen local minimum in the DEM and to iteratively increase the water level until this spill-over point is reached.

In the case where evaporation balances inflow, such as for several large present-day inland seas, the water level should be increased iteratively until the integrated evaporation over the lake matches the inflowing water flux.

A pseudo-code example of this algorithm is illustrated below.

```
water level = 0

WHILE (threshold is not reached)
  increase water level
  perform flood-fill
END WHILE
```

The threshold condition can be the overflowing of the lake into the sea, the total evaporation over the lake balancing total water influx, or any other logical condition.

Depending on the vertical resolution of the DEM and the accuracy required to determine the lake depth, such a calculation requires dozens to hundreds of iterations. This is not always a restriction for practical applications where one is interested in the water runoff pathways and drainage basins for a given DEM. However, when performing an ensemble of simulations, or a simulation where the topography changes over time, reducing the computation time can be crucial for the performance and feasibility of the application and optimizations are required. In the next paragraph we present several optimizations to the standard flood-fill algorithm as described above, which reduce computation time considerably for map with a large area or a high resolution.

## 2.2 Optimizations

### 2.2.1 Low-resolution block inspection

When filling a depression to a pre-determined water level, we strongly reduce the number of operations by starting with creating a lower resolution "maximum topography" map. For example, any element of a 4 km maximum topography map will contain the highest value of the corresponding 4 x 4 block of 1 km elements. Consequently, we apply the flood-fill algorithm to this lower resolution map. If the highest element of a 4 km x 4 km square lies below the water level, all sixteen 1 km elements must do as well, implying all of them can be flooded at once. This step yields a first coarse filling scheme without testing each individual element at the final higher resolution.

When this part of the algorithm is finished, some elements may not yet be flooded whereas they should be, but the large majority of them will be filled already if the distribution is not too scattered. To identify those elements that still need to be flooded, we take this intermediate map and stack from the 4 km fill, and use these as a starting point for a 1 km fill. Applied to the 1 km map, the algorithm will only have to fill in the "fringes" of the depression - a much smaller area than the part already processed at low-resolution. Figure 2 illustrates this.

In Fig. 2, the 1 km true/false-mask is shown as a black overlay in all panels. The 4 km stack (light green) is initiated with a seed in the lower left corner of the map, panel A. The 4 km algorithm expands from this seed until no more elements can be added. The final map (blue) and stack (light green) are shown in panel B. These are then converted to their 1 km equivalent, shown in panel C. As can be seen, the conversion conserves the filled elements. The 1 km algorithm then continues from this map and stack until no more elements can be added. The resulting map and stack coincides with the 1 km true/false-mask in panel D. The decrease in computation time resulting from using the high-resolution algorithm is larger than the additional computation time required to create the low-resolution map and to convert the low-resolution stack to its high-resolution equivalent. This is shown for an example in Sect. 3.1.

A further increase in efficiency can be achieved by creating an additional medium-resolution map in between. For example, we can start with a 10 km fill, and then convert the resulting map and stack to 5 km resolution. Use the intermediate result for a 5 km fill algorithm, run this until completion, convert the resulting map and stack to their 1 km equivalents, and finalize with the 1 km fill. Note that this will only work when the different resolutions are integer multiples of each other. Constructing rules for when this condition is not met are not evident, and will increase computation time, and are beyond the scope of this study.

A Matlab implementation of the fill algorithm is provided in the supplementary material, being script "Demo_FillSea.m".

### 2.2.2 Shoreline memory

In the case of a depression that must be filled up to overflow conditions, one small improvement to the flood-fill algorithm is straightforward. Any element that is filled at a certain water depth will also be filled for any higher water depth, implying that the lake's shoreline will only expand outward when the water depth is increased. Hence, in the improved flood-fill

algorithm, a stack element that is found to lie above the water level is not removed from the stack. Instead it is flagged and not inspected again in any further loops. This means that when the algorithm is finished, the stack contains all those and only those elements, which directly border filled elements but which are not filled, being the shoreline.

When the water depth is increased during the filling of a depression, the final stack from the previous loop is taken as a starting point. This means that all elements of the final, deepest lake have only been inspected once during the iteration procedure. For a deep lake, which requires many depth increments to be filled, the decrease in the required computation time can be large given this approach, as will be shown in Sect. 3.3.

**2.2.2 Low-resolution lake depth estimation**

When trying to fill a depression up to the level of overflowing, i.e. with no pre-determined water level, it is generally not possible to use the low-resolution block inspection algorithm presented in Sect. 2.2.1. This is because the algorithm needs to check if overflow is reached for every single depth increment, which has to be done at 1 km resolution. Because it is not possible to convert the high-resolution stack and map back to the lower resolution, we cannot use the final high-resolution shoreline to start a new low-resolution fill. It is therefore necessary to perform all fill iteration at high resolution, which is computationally expensive.

Performing a fill on a low-resolution average topography map solves this issue. Although not mathematically necessary, in practise the topography of a geographical area is usually smooth enough to yield a low-resolution water depth estimate close to the "true" water depth that would be calculated with a high-resolution fill. This means that, instead of starting at zero depth, we can initiate the algorithm with a depth slightly below the depth yielded by the low-resolution estimate. The lake at this depth will be close to its maximum extent, so we can use the block inspection method to efficiently fill the majority of the lake's central elements at low resolution, thus reducing the computation time. The lake's fringes are filled with the high-resolution algorithm and the depth is increased incrementally until overflow is reached. Section 3.4 describes an experiment where this method is implemented. A pseudo-code example of this algorithm is illustrated below.

```
water level = 0

WHILE (threshold is not reached)
  increase water level
  perform flood-fill on 40 km average topography field
END WHILE

decrease water level
perform flood-fill on 40 km maximum topography field
convert 40 km map and stack to 1 km equivalent

WHILE (threshold is not reached)
  increase water level
  perform flood-fill on 1 km topography field
END WHILE
```

A Matlab implementation of this algorithm is provided in the supplementary material, being script "Demo_FillLake.m".

## 3 Results

All experiments are performed in Matlab R2014b on a 2013 iMac with a 3.2GHz Intel i5 processor and 8GB 1600MHz
DDR3 Memory. Note that Matlab is an interpreted language and that the performance of the algorithm in a compiled
language such as Fortran or C will generally be much faster. However, the relative improvements in performance of the
optimized algorithms should be preserved.

### 3.1 Low-resolution block inspection

As a first example, we use different versions of the flood-fill algorithm to determine the land/ocean-mask for a DEM of
North America and Greenland (Amante and Eakins, 2009), using an oblique stereographic projection at 1 km resolution. The
region covers an area of roughly 7000 by 5000 km, resulting in approximately 17 million ocean elements to be filled. This
extremely large size is useful for testing the efficiency of different set-ups and is needed in order to model the pressure
exerted by lakes in gravitationally consistent calculations during the evolution of ice-sheets in North America.

The chosen horizontal resolution of 1 km is very high for such models. However, a lower resolution would overlook
topographical features, such as small river valleys, that would limit the water level of the lake through drainage. This would
lead to a systematic overestimation of the total water volume unrelated to the uncertainty of the modelled location of the ice
sheet margin, which is usually not as high as 1 km. A comparison between the calculated water volume of a lake at 1 km
resolution versus the calculated volume of the same lake at 40 km resolution showed that the difference can be as large as 20
%.

Creating the land/ocean mask depicted in Fig. 3 with the standard version of the flood-fill algorithm at a 1 km resolution
takes approximately 82 seconds. This serves as benchmark to which all subsequent experiments will be compared.

We performed two series of experiments with block inspection at different resolutions. The first series considers a sequence
of 40 km, 8 km, 4 km, 2 km and 1 km resolutions. The second series considers a sequence of 40 km, 20 km, 10 km, 5 km
and 1 km resolutions. The results in terms of computational efficiency are presented in Table 1 and Table 2, respectively.
Both the conversion and filling parts of the Tables can be read as "coming from [row] resolution, going to [column]
resolution". For example (Table 1; highlighted in yellow), converting a 40 km map and stack to their 1 km equivalent takes
0.20 seconds. A 1 km fill starting with that map and stack will then take 27.54 seconds. The diagonal elements in the filling
parts of the tables indicate the computation time of a fill at that resolution starting from a single seed element.

An overview of several possible resolution schemes is given in Table 3. It shows that the most efficient scheme (40 km - 5 km - 1 km) is almost an order of magnitude faster than the default algorithm at 1 km resolution.

## 3.2 Shape dependence

The total gain in efficiency from the improvements to the algorithm will of course depend on the details of the problem it is applied to. In the optimized algorithm, the number of elements that can be filled at the low resolution roughly scales with the area of the depression, whereas the number of elements that need to be filled at the high resolution roughly scales with the depression's circumference. The gain in computational efficiency therefore depends largely on the shape of the depression, becoming larger when the depression becomes more circular.

To illustrate this, the improved algorithm with a 40 km - 5 km - 1 km resolution scheme was applied to several versions of the same topography field, passed through increasingly strong high-pass filters with a cut-off wavelength of 30km, so that the remaining topographical features that determine the shape of the depression are all too small to be visible on the low-resolution map in Fig. 4.

The results of the experiment are given in Table 4. The area of the ocean basin and therefore the number of elements that require filling does not change much. However, the compactness factor C of the ocean basin, which is defined as:

$$C = \frac{A}{4\pi\emptyset^2}$$

with $\emptyset$ the circumference and A the area of the basin, decreases sharply when the small-scale topographical features become more prominent.

This illustrates that the gain in computational efficiency from the block inspection depends strongly on the shape of the topography in which a depression must be filled. Natural topographical features generally have a vertical scale proportional to their horizontal scale, which leads to the relatively circular shapes of most natural lakes. This means that, for most natural topographies, the block inspection algorithm will substantially decrease the computation time required for filling depressions.

## 3.3 Shoreline memory

Similar to the previous experiment, we consider the lake formation in the North American region 30,000 years ago as a second example. At this time, large parts of the North American continent were covered by the Laurentide ice-sheet. The depression left in the bedrock by the weight of the ice, combined with the mass of ice damming off the Hudson Strait lead to the formation of a massive proglacial lake over the area of what is now known as the Hudson Bay. The situation is shown in Fig. 5.

We apply the flood-fill algorithm to the same 1 km resolution topographic map, combined with an ice thickness distribution, a bedrock deformation field and a geoid perturbation field (all generated by the ANICE-SELEN model - de Boer et al, 2014) to determine the size of the lake. These data fields have a 40 km resolution and are used to perturb the 1 km resolution DEM,

such that the small topographic features are preserved. The 40 km resolution perturbation fields are mapped onto the 1 km resolution data fields using bilinear interpolation. The lake is found to cover an area of $3.5 \times 10^6$ km$^2$, with a volume of $6.0 \times 10^{14}$ m$^3$ or about 1.7 meters eustatic sea-level equivalent of water.

As a benchmark, we apply the standard flood-fill algorithm without any optimizations, using a 5 m depth increment. This means that the algorithm fills the depression up to a certain water level, increases the depth by 5 m and fills the depression up to the new water level from the same seed, until the lake overflows into the sea. This run costs 351 seconds (almost six minutes) to complete. This implies that for a 120,000 year period which is resolved every 5 years, which is a time step typically used to resolve the last glacial cycle in an ice-sheet model, computational time is excessive relative to the approximately 50 hours such a simulation would otherwise require.

Performing the same test with a depth increment of 10 m takes 179 seconds. This is far less because for each depth interval the complete lake has to be filled, so the number of fills increases more or less linearly with the requested accuracy. A 5 m depth interval leads to an accuracy of about 3 % in the lake's volume, which is considered to be sufficiently accurate.

By implementing the "shoreline memory" improvement described in 2.2.2 to the algorithm we improve the performance in terms of computation time. As explained earlier, the improved shoreline memory implies that every element will only be checked once, yielding a drastic reduction in computation time, which is particular relevant for large lakes. Filling the same lake at 1 km resolution only takes 23 seconds in the optimized case, a reduction in computation time of about 90 % with respect to the benchmark.

### 3.4 Low-resolution lake depth estimation

As a final improvement, we implement the depth estimation method. This method starts by filling the same lake on the 40 km resolution topography, in order to get a depth estimate. The resulting lake is depicted in Fig. 5. It has a volume of $6.6 \times 10^{14}$ m$^3$, overestimating the "true" 1 km volume by about 10 %. The 40 km estimate covers several river valleys visible in Fig. 6c, which should drain the lake. Since these valleys are only a few kilometres wide they do not appear on the smoothed 40 km resolution topography map. This causes the overestimation of the water level and the lake's extent and volume - hence a high resolution is needed. The resulting depth is reduced by an arbitrary amount of 20 % and a fill to this new depth is performed on the 40 km maximum topography field. The resulting lake, which will serve as a starting point for the final 1 km fill, is depicted in Fig. 6b.

The water level is then incrementally increased, while using the 1 km fill algorithm, until the lake overflows into the sea. The complete process takes 4.4 seconds to complete. This is about 5 times faster than the 1 km fill with shoreline memory and about 80 times faster than the default implementation, a reduction in computation time of about 99 % with respect to the benchmark experiment.

Note that the 20 % reduction in depth from the 40 km estimate is chosen somewhat arbitrarily. Depending on the smoothness of the local topography, it is possible that the 40 km estimate will be much more accurate. In such cases, it is justifiable to

reduce the depth less, and hence improve efficiency even more. For this reason, it is advisable to check if the depth reduction from the 40 km estimate is enough to prevent overflow. If not, the depth needs to be increased even further before starting the 40 km maximum topography fill.

**3.4 Comparison to the drainage-pointer method**

5 As a benchmark of our method, we compare the computational performance of our optimized algorithm to that of the commonly used drainage-pointer approach (Tarasov and Peltier, 2006). This method assigns to every map element a drainage pointer, indicating which one of its immediate neighbours receives run-off from that element. By following the run-off from an element until it reaches the sea, elements can be grouped together by the location where run-off reaches the sea, thus creating drainage basins.

We found that a Matlab implementation of the first step of this approach, where drainage pointers are assigned to all high resolution elements, required about five times more computation time than our optimized algorithm when used for determining the land/ocean mask as in section 3.1. The reason for this is that the drainage-pointer approach needs to assign these pointers to every single high resolution map element in order to work, whereas our method only needs to treat those

15 elements that are beneath the water level and then only those near the shoreline at high resolution, thus limiting the number of operations.

**4 Conclusions**

We have presented and evaluated several optimizations to a standard flood-fill algorithm. When determining the land/ocean-mask over a large grid, the optimized algorithm is up to 9 times faster than the default algorithm. When determining the

20 extent and depth of Lake Agassiz from a given DEM and ice thickness map, the optimized algorithm is even up to 80 times faster than the default algorithm as proposed by Zhu et al. (2006). The gain in computational efficiency depends on the smoothness of the topography, with the largest reduction in computation time achieved when most of the volume of the depression is contributed by topographical features with a horizontal scale larger than the lowest resolution of the block inspection method, as has been explained in section 2.2.2.

In the example given in this study, the ice thickness, bedrock deformation and geoid anomaly all initially have a 40 km resolution. In order to do a 1 km lake fill, these fields were interpolated onto a 1 km grid, which is computationally expensive (about 11 seconds in our Matlab implementation). If all input fields are already initially available at high resolution, they only need to be downscaled to a low resolution for the block inspection step, which takes considerably less

time. For this reason, the computation time for this interpolation step is not included in the results.

The algorithm presented takes into account the bedrock deformation and geoid perturbation caused by the mass changes of an ice-sheet, if they have been calculated. These effects both influence the volume of a lake by deepening the basin because of the vertical water pressure on the landscape and by raising the water surface.

Any study involving either a very large grid or a large number of repeated simulations will greatly benefit from these optimizations. Examples include the modelling of large proglacial lakes, the dynamical modelling of run-off over an evolving ice-sheet, changes of water routing by tectonic activity and changes in a large water basin due to sea-level changes such as in the Mediterranean.

**Code and data availability**

Several Matlab scripts containing different versions of the flood-fill algorithm, as referenced in this manuscript, are provided as supplementary material. Two large NetCDF files containing data files required to run these scripts are available online at doi:10.5194/gmd-2016-85-supplement, or directly at https://zenodo.org/record/49614.

**Acknowledgements**

The Ministry of Education, Culture and Science (OCW), in the Netherlands, provided financial support for this study
via the program of the Netherlands Earth System Science Centre (NESSC). Heiko Goelzer, Lennert Stap and Sarah Bradley commented on an earlier draft of this paper.

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

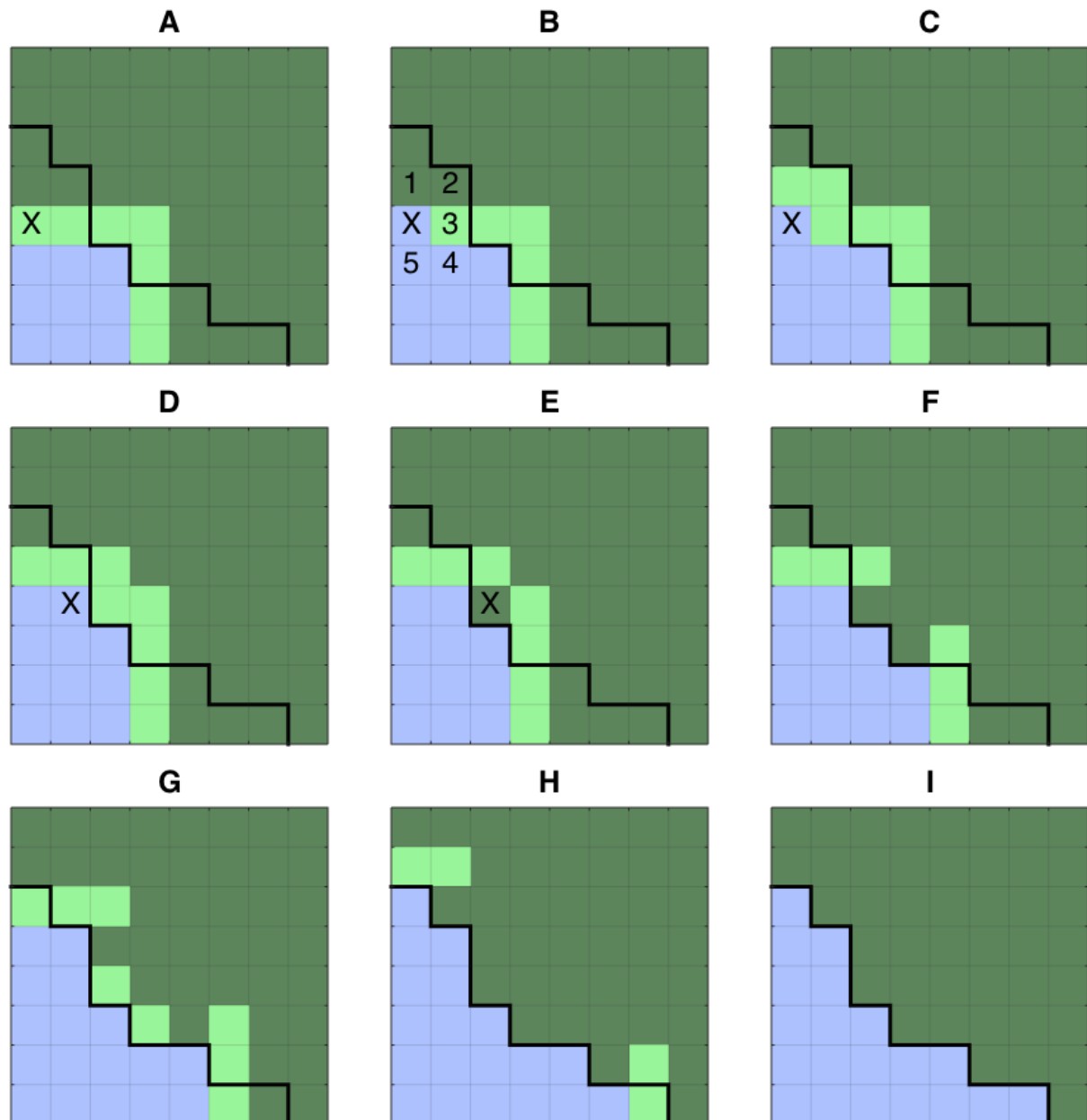

**Figure 1: An illustration of the default flood-fill algorithm. Dark green: unfilled; light green: stack; blue: filled. Black line: shoreline. A: the algorithm starts at the first element in the stack (marked with X). B: the element is found to meet the fill criterion and is filled. Its immediate neighbours (marked 1 to 5) are now inspected. Neighbours 1 and 2 are not yet in the stack and so they are added. Neighbour 3 is already in the stack and neighbours 4 and 5 are already filled. C: result of checking the five neighbours. D: The algorithm now moves on the next element in the stack (marked with X), which is consequently filled and its top-right neighbour is added to the stack. E: The next element in the stack is found not to meet the fill criterion (since it lies above sea level) and so it is removed from the stack. F: The result after inspecting the whole stack once. G: The result after inspecting the whole stack again. H: The result after inspecting the whole stack again. I: The final result.**

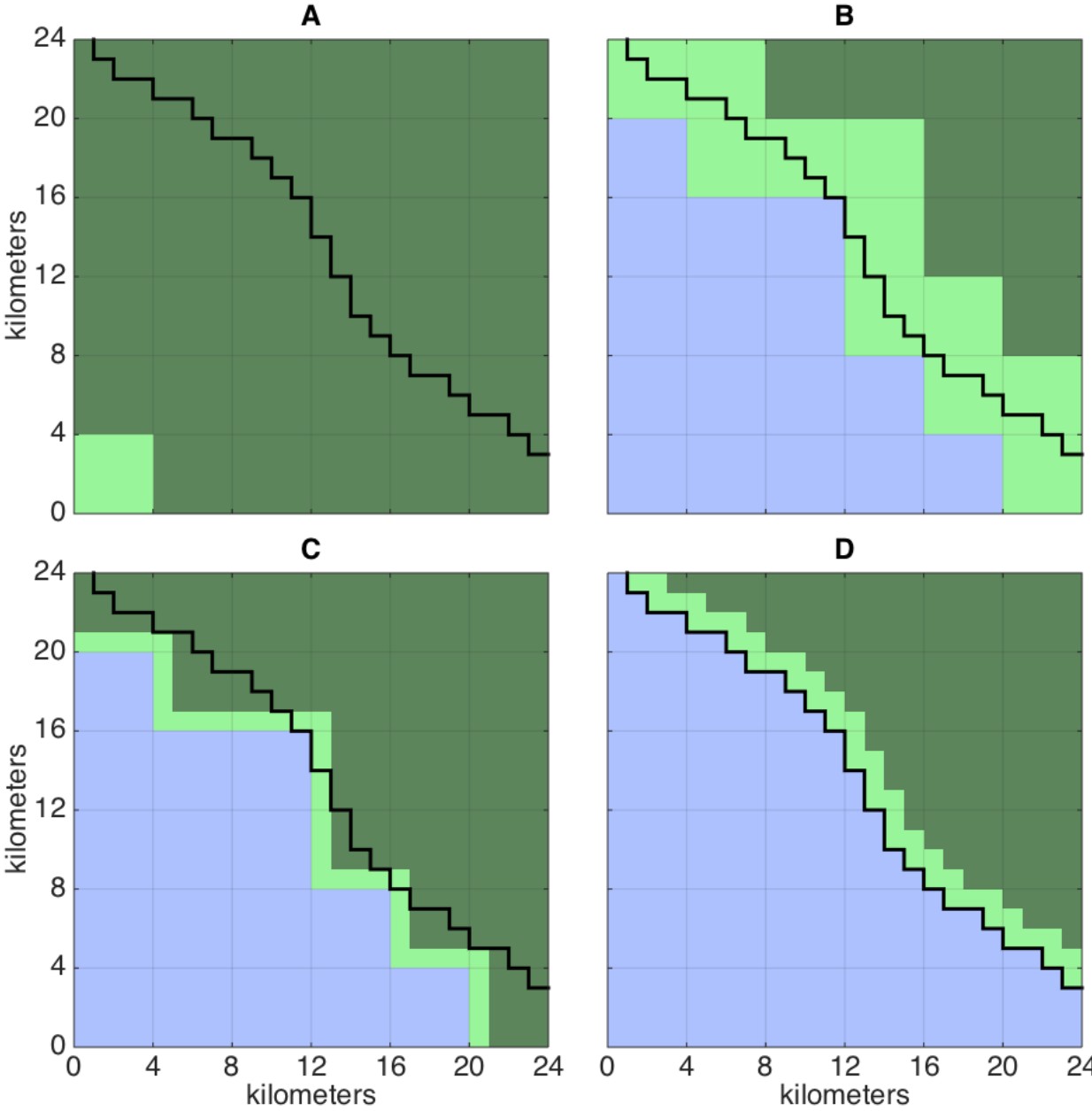

**Figure 2: An illustration of the low-resolution block inspection improvement upon the default flood-fill algorithm. Dark green: unfilled; light green: stack; blue: filled. Black line: shoreline at 1 km resolution. A: the 4 km fill is given a seed in the southwest corner of the map. B: the 4 km fill is completed. C: The 1 km map and edge created from the intermediate 4 km output. D: the result of the 1 km fill.**

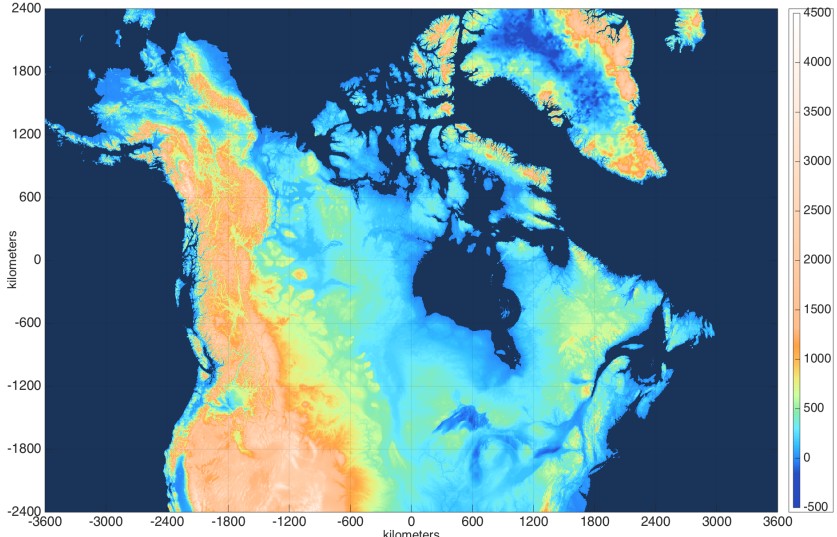

**Figure 3: The present-day land/ocean-mask (uniform deep blue) and the bedrock topography at 1 km resolution.**

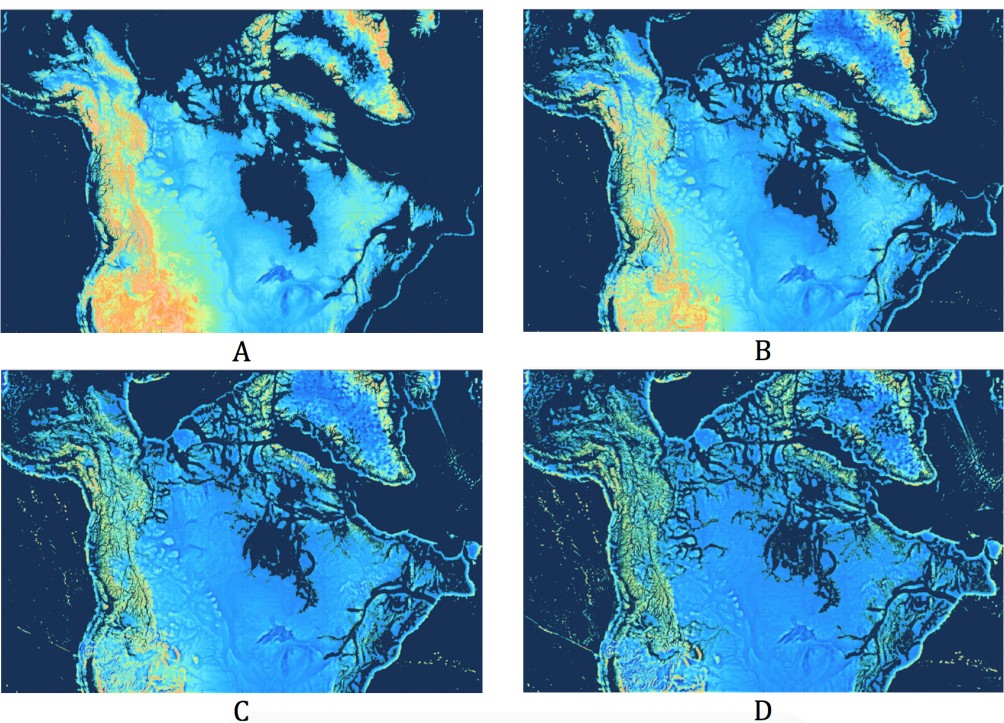

**Figure 4: Land/ocean-mask for the same region, with the topography passed through a high-pass filter. Panel A: 40 % reduction in amplitude for wavelengths above 30km. Panel B: 60 % reduction. Panel C: 80 % reduction. Panel D: 90 % reduction. The ocean area increases slightly and the length of the coastline increases strongly.**

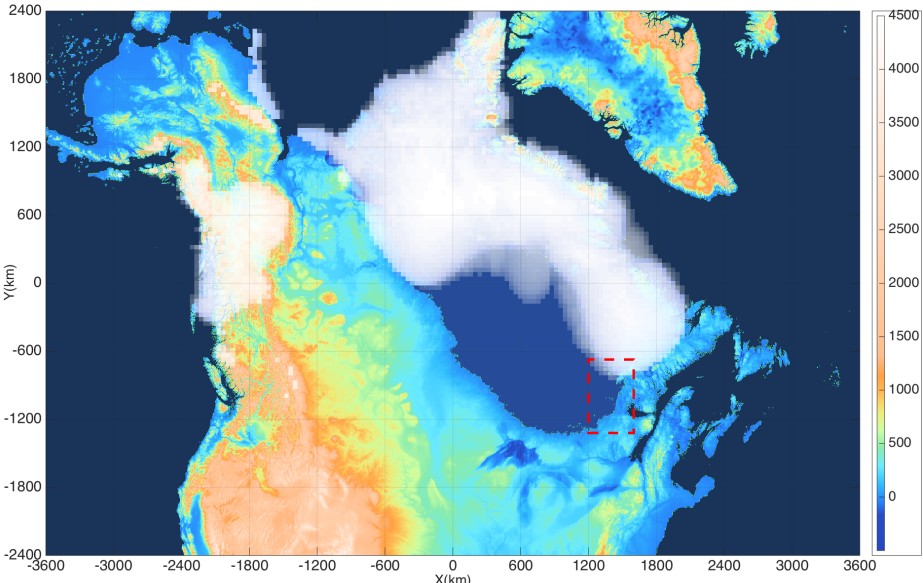

**Figure 5: North America, the Laurentide ice-sheet and Lake Agassiz, 30ky before present. The red outlined region is shown in close-up in Fig. 6.**

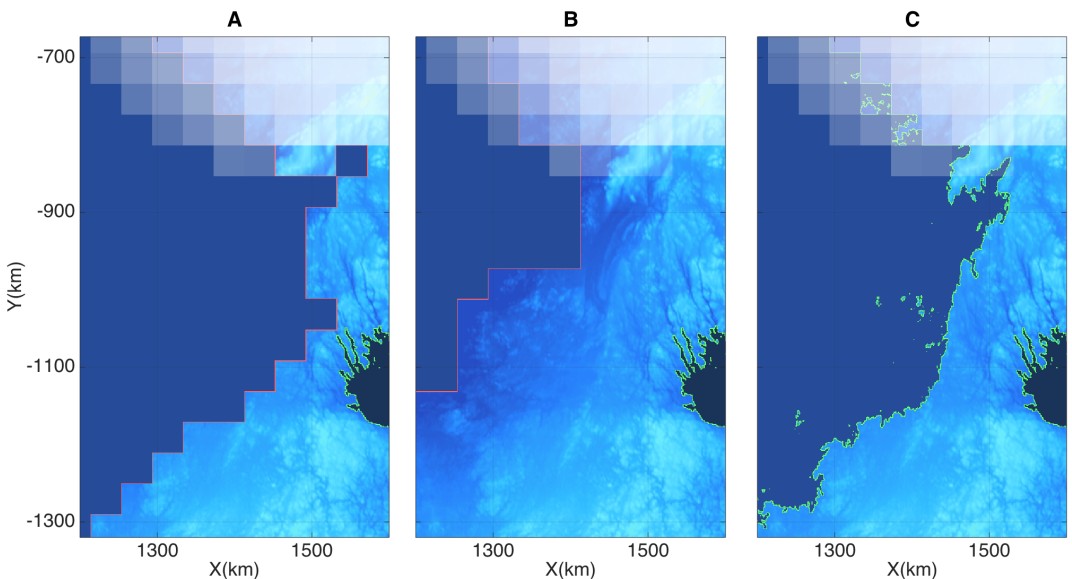

**Figure 6: The area around the eastern shore of the lake, outlined in red in Fig. 5. Panel A: The 40 km resolution estimate. Panel B: The result of the 40 km maximum topography fill to a water depth 20 % below that of the 40 km estimate. Panel C: The final 1 km lake. Note that the algorithm allows for ice to float when the water column is high enough, creating a small ice-shelf along the margin, visible in panel C where part of the lake's shoreline runs beneath the ice.**

**Table 1: Computation time in seconds for stack and map conversion between different resolutions and for flood-filling at different resolutions with different starting stacks, for the 40 km - 8 km - 4 km - 2 km - 1 km series. The yellow shaded numbers are explained in the text.**

| Stack and map conversion | | | | | | Flood-fill | | | | |
|---|---|---|---|---|---|---|---|---|---|---|
| | **40** | **8** | **4** | **2** | **1** | **40** | **8** | **4** | **2** | **1** |
| **40** | - | 0.04 | 0.04 | 0.07 | 0.20 | 0.05 | 0.19 | 0.85 | 3.73 | 27.54 |
| **8** | - | - | 0.34 | 0.37 | 0.56 | - | 0.94 | 0.16 | 0.76 | 11.65 |
| **4** | - | - | - | 1.30 | 31.22 | - | - | 4.03 | 0.40 | 8.29 |
| **2** | - | - | - | - | 65.98 | - | - | - | 16.89 | 5.64 |
| **1** | - | - | - | - | - | - | - | - | - | 82.04 |

**Table 2: Computation time in seconds for stack and map conversion between different resolutions and for flood-filling at different resolutions with different starting stacks, for the 40 km - 20 km - 10 km - 5 km - 1 km series.**

| Stack and map conversion | | | | | | Flood-fill | | | | |
|---|---|---|---|---|---|---|---|---|---|---|
| | **40** | **20** | **10** | **5** | **1** | **40** | **20** | **10** | **5** | **1** |
| **40** | - | 0.04 | 0.03 | 0.05 | 0.21 | 0.06 | 0.02 | 0.05 | 0.18 | 13.78 |
| **20** | - | - | 0.08 | 0.08 | 0.34 | - | 0.17 | 0.02 | 0.12 | 11.26 |
| **10** | - | - | - | 0.24 | 0.43 | - | - | 0.62 | 0.08 | 9.28 |
| **5** | - | - | - | - | 1.07 | - | - | - | 2.45 | 7.97 |
| **1** | - | - | - | - | - | - | - | - | - | 82.11 |

**Table 3: Total computation time relative to the 1 km benchmark experiment for several resolution schemes, sorted ascending.**

| Resolution scheme | | | | | Computation time reduction (%) |
|---|---|---|---|---|---|
| 40 | - | - | 5 | 1 | 88.64 |
| 40 | 8 | - | - | 1 | 84.79 |
| - | 8 | 4 | - | 1 | 50.13 |
| - | 8 | - | 2 | 1 | 10.25 |
| - | - | 4 | 2 | 1 | 5.80 |
| - | - | - | | 1 | 0.00 |

**Table 4: Reduction in computation time for the 40 km - 5 km - 1 km resolution scheme with respect to the 1 km benchmark experiment for different bedrock topographies. Also listed are the area and compactness factor of the resulting ocean basin.**

| Filter (%) | Compactness factor | Ocean area ($km^2$) | Computation time reduction (%) |
|---|---|---|---|
| -100* | 6.0e-2 | 1.72e7 | 95.48 |
| 0 | 5.1e-3 | 1.72e7 | 91.56 |
| 20 | 2.7e-3 | 1.76e7 | 89.37 |
| 40 | 1.4e-3 | 1.77e7 | 74.48 |
| 60 | 7.4e-4 | 1.76e7 | 58.89 |
| 80 | 2.6e-4 | 1.79e7 | 35.35 |
| 90 | 1.4e-4 | 1.86e7 | 18.72 |

*30km low-pass filter; "smoothed" topography.