# Peer review of "A computationally efficient depression-filling algorithm for digital elevation models applied to proglacial lake drainage"

_Geoscientific Model Development, 2016_

## Referee Comment (RC1) · L. Tarasov (Referee) · 29 Aug 2016

This submission offers 3 relatively simple ways to make depression (ie lake) filling in surface drainage solvers significantly more computationally efficient at O(1 km) horizontal resolutions. With current efforts by a number of modelling groups to fully couple ice sheet and climate models, this is a topical issue.

This study contrasts with my approach, briefly described in T & P 2006 that focussed on coarsening the hydrological DEM resolution to the resolution of the ice sheet grid while preserving routing pathways. It would be worth a few sentences comparing the

two approaches with respect to computational speed and accuracy given the different tradeoffs between the two approaches and the contextual accuracy of the ice margin. The last point needs to be underlined as the uncertainties in paleo ice sheet margins will always be much larger than 1 km (and I don't see 1 km grid resolution continental scale ice sheet models running glacial cycles anytime soon). Heck, there are few locations along the Laurentide ice sheet where we will confidently never know the ice margin location to even +/- 40 km resolution at any given time (barring some new dating technique).

Once the few specific comments below are addressed, this paper does deliver some significant improvements to lake filling algorithms on high-resolution grids, and as such is worth publishing in GMD.

**specific comments**

:: 29 Tarasov and Peltier, 2004).

**inappropriate reference, should be Tarasov and Peltier, 2005 and 2006**

:: Lake Agassiz.... 6 It is therefore important to accurately model the extent and volume of the lake over time

**Tarasov and Peltier, 2006 would I think be a relevant reference for this since they model Lake Agassiz (other other North Am pro-glacial lake) evolution**

::the largest of which is Lake Agassiz, along the southern margin of the ice-sheet. Lake Agassiz ... :: Doing this requires an accurate treatment of the large changes in the land/ocean-mask that occur where the ice-sheet covers most of the Canadian Arctic Archipelago and blocks the Hudson Strait. This changes the location where lake outflow reaches the sea over time

**The above is geographically/geologically incorrect and has no relevance to Lake Agassiz. Neither the Canadian Arctic Archipelago (CAA) nor Hudson Strait ice were drainage blocks for Lake Agassiz. The possible northern drainage outlet for Lake**

**[GMDD](https://...)**

Agassiz is the Mackenzie River delta which is outside of the CAA. Ice across Hudson Bay and Northern Ontario is what dammed the lake in the direction of Hudson Strait drainage according to consensus geological inferences (cf eg, Dyke, 2004). And the 8.2 ka (not 8.4) drainage was for proglacial Lake Ojibway not Agassiz.

:: we consider the lake formation in the North American region 30,000 years ago as a second example. At this time, large parts of the North American continent were covered by the Laurentide ice-sheet. The depression left in the bedrock by the weight of the ice, combined with the mass of ice damming off the Hudson Strait lead to the formation of a massive proglacial lake over the area of what is now known as the Hudson Bay :: and figure 4 # Should make clear that this lake and ice configuration is from your model and has no geological validation for that time (or if it does, then do provide the relevant citation)

:: In the example given in this study, the ice thickness, bedrock deformation and geoid anomaly all initially had a 40 km resolution. In order to do a 1 km lake fill, these fields were interpolated onto a 1 km grid, which is computationally expensive. If all input fields are already at high resolution, they only need to be downscaled to a low resolution for the block inspection step, which takes considerably less time. For this reason, the computation time for this interpolation step is not included in the results

**Given that no one will be running 1km grid resolution ice sheet models for glacial cycle contexts (given "proglacial" in the title) in the foreseeable future, give the interpolation time to provide a complete time budget.**

:: Supplement

I have not been able to test the code since the required netcdf files are not on this server. But I have a few suggestions:

1) the ReadMe.txt should provide command line examples how to run the scripts (so that the reader doesn't have to dig right away into the code to see if there are any

arguments that need passing).

2) Verify that the code runs on octave. What is the point of using open source publishing to publish something that requires a close source app especially when an open source alternative is available?

3) add the required net-cdf files for the sample scripts on the GMD page (as a separate supplement...)

---

## Referee Comment (RC2) · Anonymous Referee #2 · 5 Sep 2016

The paper presents a number of efficiency improvements to an algorithm used to quantify lake depths in topography. The improvements lead to a much reduced computational time, useful for many iterations over glacial time periods.

While the work carried out looks to be robust and of value to a fairly specific application, I wonder whether the work has a broad enough reach to be published in a relatively high impact journal such as GMD. The first line of the paper suggests that the problem of determining lake depths is often encountered in many fields, but doesn't go on to give any examples beyond proglacial lakes. And do these other applications need to

solve this problem over many iterations? Otherwise the computational time of minutes is not a massive issue, and the techniques presented have a fairly niche application, and could be presented in a paper specific to the application.

The introduction does not give enough background to the task, such that I had to read it a number of times to understand exactly what the task at hand was. Further there is not much detail on the default algorithm, the reader is left to go and read the paper by Zhu et al. (2006), such that it is a bit hard to follow independently.

There are also a lot of vague statements such as "A second issue is an efficient way to determine the water level" on page 3, line 22. Because you don't introduce your terms, such as "base level" and "water level" it is not always easy to follow the methods. Be more specific in these statements to help the reader understand what it is going on. Another example, on page 4, line 18 and Fig. 1, you talk about a true/false mask but don't define what this is. In Fig. 2 you talk about a land/ocean mask, but all I can see is DEM elevations? Is the deep blue actually a mask and not elevations? In this case, this needs to be in the legend.

In summary, the paper looks to be a useful study, but if it is to be accepted must give much more background to the topic and the previous methodology and explain the work being carried out much more clearly.

If the Editor feels the topic is justified being a stand-alone paper in GMD, then with major revisions, the paper could be publishable, however, the authors need to justify the applicability of their improvements beyond their specific case study.

---

## Referee Comment (RC3) · L. Tarasov (Referee) · 30 Sep 2016

**reviewer comment: This study contrasts with my approach, briefly described in T P 2006 that focused on coarsening the hydrological DEM resolution to the resolution of the ice sheet grid while preserving routing pathways. It would be worth a few sentences comparing the two approaches with respect to computational speed and accuracy given the different tradeoffs between the two approaches and the contextual accuracy of the ice margin.**

**author response: We agree that a comparison of the two methods in terms of computational speed is of added value to the manuscript. Although we don't have the code from T P 2006, we worked along the concepts of their algorithm during the start of our project, but quickly concluded that this approach was computationally more expensive. This is mainly because the drainage pointer approach must be applied to the whole region, meaning that, although it has a larger scope, it needs to operate on every grid cell. Our approach only treats the flooded grid cells of a designated drainage basin. For the case considered our code is a factor 5 faster. This will be described in a separate section in the manuscript**

**Reviewer response to above: You've lost me. My model computes mean (ie over past 100 years) surface drainage every 100 years over a glacial cycle with less than a 30 minute computational overhead (for surface drainage) for the whole run. Is your approach really 5 times faster than this? You seem to be addressing my drainage pointer algorithm instead of my approach of surface DEM upscaling that preserves drainage routing (with an accuracy displayed in the original 2006 QSR paper).**

---

## Author Comment (AC1) · 30 Sep 2016

We like to thank the reviewer for his comments on the manuscript and would hereby like to address the concerns he raised.

In Italics the comments, below our rebuttal

*This study contrasts with my approach, briefly described in T P 2006 that focused on coarsening the hydrological DEM resolution to the resolution of the ice sheet grid while preserving routing pathways. It would be worth a few sentences comparing the two approaches with respect to computational speed and accuracy given the different*

*tradeoffs between the two approaches and the contextual accuracy of the ice margin.*

We agree that a comparison of the two methods in terms of computational speed is of added value to the manuscript. Although we don't have the code from T P 2006, we worked along the concepts of their algorithm during the start of our project, but quickly concluded that this approach was computationally more expensive. This is mainly because the drainage pointer approach must be applied to the whole region, meaning that, although it has a larger scope, it needs to operate on every grid cell. Our approach only treats the flooded grid cells of a designated drainage basin. For the case considered our code is a factor 5 faster. This will be described in a separate section in the manuscript

*The last point needs to be underlined as the uncertainties in paleo ice sheet margins will always be much larger than 1 km (and I don't see 1 km grid resolution continental scale ice sheet models running glacial cycles anytime soon). Heck, there are few locations along the Laurentide ice sheet where we will confidently never know the ice margin location to even + 40 km resolution at any given time barring some new dating technique)*

We agree that the uncertainty in ice margin reconstructions will likely never reach the 1 km resolution described in our manuscript, at least not everywhere. However, we believe that this does not detract from the added value of a 1 km lake reconstruction over a 40 km version. Most topographical features that would limit the water level of such a lake through draining, such as river valleys, have horizontal dimensions that are far smaller than 40 km, meaning that a 40 km analysis would overlook these features and thereby overestimate the water volume. Hence solving at 40 km introduces a systematic error, which is only partly related to the uncertainty in the location of the ice margin. A 1 km analysis strongly reduces this systematic error. We will add a few sentences to the "Introduction" section of the manuscript to clarify this point and quantify the difference in calculated water volume.

*:: 29 Tarasov and Peltier, 2004).*
*inappropriate reference, should be Tarasov and Peltier, 2005 and 2006*

We apologize for this erroneous reference and will correct this.

*:: Lake Agassiz.... 6 It is therefore important to accurately model the extent and volume of the lake over time*
*Tarasov and Peltier, 2006 would I think be a relevant reference for this since they model Lake Agassiz (other North Am pro-glacial lake) evolution*

We agree that this is a relevant reference and will add this to the manuscript.

*::the largest of which is Lake Agassiz, along the southern margin of the ice-sheet. Lake Agassiz ... :: Doing this requires an accurate treatment of the large changes in the land-mask that occur where the ice-sheet covers most of the Canadian Arctic Archipelago and blocks the Hudson Strait. This changes the location where lake outflow reaches the sea over time*
*The above is geographicallyincorrect and has no relevance to Lake Agassiz. Neither the Canadian Arctic Archipelago (CAA) nor Hudson Strait ice were drainage blocks for Lake Agassiz. The possible northern drainage outlet for Lake Agassiz is the Mackenzie River delta which is outside of the CAA. Ice across Hudson Bay and Northern Ontario is what dammed the lake in the direction of Hudson Strait drainage according to con-sensus geological inferences (cf eg, Dyke, 2004). And the 8.2 ka (not 8.4) drainage was for proglacial Lake Ojibway not Agassiz.*

We agree that the manuscript may be confusing here - indeed, we do not wish to suggest that any drainage events happened through the CAA. However, our manuscript does not concern any particular drainage event. It proposes a mathematical algorithm, which can be used as a tool to study such events. We have chosen a model-generated ice sheet configuration, which allows for a (perhaps unrealistically) large proglacial lake to form, because this creates the most computationally expensive setting, and therefore optimally illustrates the advantages of our approach - a configuration, which indeed

may never have existed in reality. We will clarify this reasoning in the manuscript.

*Given that no one will be running 1km grid resolution ice sheet models for glacial cycle contexts (given "proglacial" in the title) in the foreseeable future, give the interpolation time to provide a complete time budget*

We agree, and will provide these numbers in the manuscript.

*: Supplement*
*I have not been able to test the code since the required netcdf files are not on this server. But I have a few suggestions:*
*1) the ReadMe.txt should provide command line examples how to run the scripts (so that the reader doesn't have to dig right away into the code to see if there are any arguments that need passing).*

We agree, and will add these examples to the ReadMe.txt file.

*2) Verify that the code runs on octave. What is the point of using open source publishing to publish something that requires a close source app especially when an open source alternative is available?*

We agree, and are currently working on this. We do not expect any trouble, since the codes only use very basic function calls, all of which (including the NetCDF package) are available in Octave. We will include functional Octave scripts in the supplementary material.

*3) add the required net-cdf files for the sample scripts on the GMD page (as a separate supplement...)*

The required NetCDF files are freely available online and can be found with their own separate DOI, which is mentioned in the "Code and data availability" section of the manuscript. However, we have indeed discovered that not all internet browsers and search engines handle DOI's equally. We will therefore add the URL for the NetCDF files.

---

## Author Comment (AC2) · 30 Sep 2016

We like to thank the reviewer for his comments on the manuscript and would hereby like to address the concerns raised.

In Italics the comments, below our rebuttal

*While the work carried out looks to be robust and of value to a fairly specific application, I wonder whether the work has a broad enough reach to be published in a relatively high impact journal such as GMD. The first line of the paper suggests that the problem of determining lake depths is often encountered in many fields, but doesn't go on to*

[Figure]

*give any examples beyond proglacial lakes. And do these other applications need to solve this problem over many iterations? Otherwise the computational time of minutes is not a massive issue, and the techniques presented have a fairly niche application, and could be presented in a paper specific to the application.*

We agree that we could have expressed the relevance of the work a bit better than we did. We provided a couple of papers with applications in hydrology for which our work has relevance (Tarboton et al., 1991, Zhu et al., 2006, Goelzer et al., 2012 and references therein), where drainage direction maps need to be determined while accurately accounting for lakes within the maps. In addition, there are studies looking at changes in global hydrology either over past climate changes or future projections (e.g. Renssen and Knoop, 2000) where the boundary conditions of the problem may change over time. While we believe that hydrological studies benefit from our improved algorithm, we agree that the main application will probably still be the treatment of proglacial lakes within ice models.

Having said that, we would like to stress that the dynamics of ice sheets is an emerging field of interest driven by the need to address the uncertainty in sea level projections and the appearance of more and more empirical data and paleoclimatological data. Meltwater pulses for instance are still poorly understood. What we hope to offer with this manuscript is a generic tool for all ice sheet models, which researchers can use to test the importance of lakes in their specific area of interest. Whether this is the Laurentide, Eurasia, the past or present and future evolution of Greenland and Antarctica is up to the individual researchers. Given the importance for hydrology and dynamical ice sheet studies we believe this model deserves a separate paper rather than an appendix to a specific application. This thought is strengthened by the fact that we need a full paper to explain the merits of the approach.

We have improved the introduction to clarify this better and added the reference to the work by Renssen and Knoop.

*The introduction does not give enough background to the task, such that I had to read it a number of times to understand exactly what the task at hand was.*

We have improved the introduction in order to clarify the problem addressed in the paper.

*Further there is not much detail on the default algorithm, the reader is left to go and read the paper by Zhu et al. (2006), such that it is a bit hard to follow independently.*

We have improved section 2 with a figure and text to illustrate how the default algorithm works as a starting point for our own configuration.

*There are also a lot of vague statements such as "A second issue is an efficient way to determine the water level" on page 3, line 22. Because you don't introduce your terms, such as "base level" and "water level" it is not always easy to follow the methods. Be more specific in these statements to help the reader understand what it is going on.*

The term "base level" in the manuscript is indeed erroneous, this should read "local topography". The term "water level" means the elevation of the water surface with respect to sea level. We clarify this in the revised version of the manuscript.

*Another example, on page 4, line 18 and Fig. 1, you talk about a true/false mask but don't define what this is. In Fig. 2 you talk about a land/ocean mask, but all I can see is DEM elevations? Is the deep blue actually a mask and not elevations? In this case, this needs to be in the legend.*

The "true/false-mask" is the same one that is introduced on page 3, line 21. We will re emphasize this in the text. Fig. 2 does indeed show the DEM elevation with the land/ocean mask as a deep blue overlay. We will clarify this in the figure caption.

---

## Author Response (AR2)

**Author comment replying to the referee comment posted by anonymous referee #2**

We like to thank the reviewer for his comments on the manuscript and would hereby like to address the concerns he raised. In Italics the comments, below our rebuttal

*The modifications made by the authors to the Introduction & Methodology are minimal in response to my comments, I do not feel that they have improved the clarity of the paper. I assume the authors believe that all the information is there so they don't think it needs modification, however, I find these parts of the paper hard to read and think the authors could be much kinder to the reader. I hoped the authors would understand my points and address them, instead I now feel I must detail*
10 *points more explicitly. I leave it to the Editor to decide whether to ask the authors to have another go at clarifying the paper for the benefit of a non-specialist.*

We apologize for having failed to adequately improve the clarity of these sections. Although we have had the manuscript read and commented upon by several colleague scientists who were not involved in the research before submission, we do
15 recognize that they may have had more relevant background knowledge than the average reader and may therefore not have picked up on some issues that can be unclear to other non-specialists. We thank the reviewer for pointing out now more specifically the difficulties with the manuscript and we have changed the manuscript accordingly.

*In the first line of the introduction, the "problem" of determining drainage routes is introduced, and the introduction then*
20 *goes on to explain the importance of the location and magnitude of the lake drainage. This is ok (though could still also be improved). However, after this, you need to be more clear about the exact problem you are addressing, why it is a problem, and how you are addressing it, and exactly what the algorithm does. On line 12, the sentence beginning "Doing this" (try and avoid using "this" - reiterate what "this" is) is not clear, and leads to confusion for me over the mask and the filling aspects. Here is my understanding of what you are doing, correct me if I am wrong...*
25 *"In order to model the extent and volume of lakes over time, and, hence, where these lakes discharge to the ocean, a land/ocean mask must first be defined. The land/ocean mask delineates ocean areas below sea level that have an open connection to the ocean, using the ice sheet extent and sea level at a given time. Large changes in the land/ocean mask occur, for example, where the ice-sheet at times covers most of the Canadian Arctic Archipelago or blocks the Hudson Strait. Alterations to the mask change the location where spill-over from the lake and drainage events reach the ocean over*
30 *time. A "flood-fill" algorithm (Zhu et al. 2006) can be used to determine the land/ocean mask with a given sea level, and then subsequently used to fill depressions that are unconnected to the ocean to a given level, or to overflow level. Quantifying the volume of the filled depressions allows the potential volume of the lake to be calculated in order to determine how much melt-water is released into the ocean and where."*

We agree that the difference between the geophysical "problem" of accounting for proglacial lakes in ice modeling, which explains the relevance of our research, and the abstract mathematical "problem" of determining the extent and volume of a depression in a DEM, which is what our algorithm solves, should be made more clear.

We have changed the description of the geophysical problem according to the reviewer's suggestion, and we have added a few lines describing the mathematical problem, and how this is solved by flood-fill algorithms in general. We have also changed the phrasing of the first paragraph of the Introduction section to avoid confusion between the two stated problems.

*Then the following clarifications: "When applying commonly used \*\*what sort of\*\* algorithms (Arnold, 2010, Doll and Lehner, 2002, Tarboton et al., 1991, Zhu et al., 2006) to a problem involving such large geographical grids and long time-scales, computation time \*\*for filling depressions?\*\* can become a limiting factor, particularly when the geometry is changing over time and the procedure has to be repeated over many time steps.*

*In this study, we describe and evaluate several improvements to a standard algorithm for filling depressions in a DEM in order to improve the computational efficiency \*\*for calculating lake volumes\*\*."*

We agree that the suggested changes aid in making the distinction between the aforementioned geophysical and mathematical problems. We have incorporated the suggested changes in the text.

*The last two paragraphs are not introduction material – these would be more suited for the methodology section.*

We agree that these two paragraphs, which justify our choice of a 1 km resolution, should be described elsewhere. We have moved them to the beginning of the Results section, after the paragraph describing the details of the ETOPO1 DEM.

*My second point: Further there is not much detail on the default algorithm, the reader is left to go and read the paper by Zhu et al. (2006), such that it is a bit hard to follow independently. I couldn't see any modification to the description, beyond the addition of a figure, which does not really help to clarify the situation very much. The description of which cells go into the stack is not written very clearly.*

We have added a few lines on the paper by Zhu et al. (2006) and more importantly improved Figure 1 and the caption, which is the one mentioned above, explaining the way the default flood-fill algorithm works. We have also changed the phrasing of the paragraph explaining the default algorithm in order to make it more readable.

*What would also help the narrative is a final line stating how you go from delineating a lake to a value for the total volume of the depression, given that (I think) that is what you have said you want to calculate in the introduction. I'm a bit confused*

*as to what you are calculating in the paper (just delineating lakes?), and what you want to calculate in the wider studies (lake volume?).*

We agree that this last step in the calculation process was missing from the explanation. We have added text to describe this.

*I also notice that in Figs. 1 and 2 that there are three green shades (and perhaps a 4th, but I think the black looks green to me!) and only two green shades described in the caption which means I find these figures confusing.*

We agree that the mentioned figures can be confusing. The problem here is that there are two "layers" of information: the state of the algorithm at a certain point in the iteration process (filled/stack elements), and the boundary conditions (the true/false-mask) under which it operates. Visualizing all this information in an intuitive way is difficult. We decided to display the true/false-mask as a semi-transparent black overlay on top of the colored blocks depicting the map and stack of the flood-fill algorithm. Two shades of green (unfilled and stack elements) combined with semi-transparent black overlay gives four possible shades of green, which leads to the confusion.

We have changed the figure, removing the overlay and instead displaying the true shoreline (the border between true and false in the true/false-mask) as a simple black line. This should improve the clarity of the figure. We have also added three additional panels depicting the results of three more full iterations of the flood-fill algorithm.

*My third point: There are also a lot of vague statements such as "A second issue is an efficient way to determine the water level" on page 3, line 22. Because you don't introduce your terms, such as "base level" and "water level" it is not always easy to follow the methods. Be more specific in these statements to help the reader understand what it is going on. Another example, on page 4, line 18 and Fig. 1, you talk about a true/false mask but don't define what this is. In Fig. 2 you talk about a land/ocean mask, but all I can see is DEM elevations? Is the deep blue actually a mask and not elevations? In this case, this needs to be in the legend.*

*Ok, the bit where you had used base level is clearer now, but I think you need to more clearly use and define different terms for the water level – sea level, lake water level and overflow level. Also, technically, how do you identify the overflow level? You use water level for all these terms, so I still find the explanation on p4 difficult to understand.*

We agree that the paragraph mentioned here contained some confusing terminology. We have rewritten the paragraph to more clearly describe how the water level of a lake is physically constrained by boundary conditions such as topography and water inflow, and how these mechanisms translate to the mathematical problem described earlier.

Based on the comments stated above, we have also made several minor changes to the rest of the Methodology and Results sections in order to improve clarity on similar subjects. We have also rewritten the abstract to more clearly place our work in its broader scientific context, in line with earlier comments of the same reviewer.

**List of changes**

10 Page 1, line 6: Rewrote the abstract to more clearly place our work in its broader scientific context, in line with earlier comments of anonymous referee #2.

Page 1, line 22: Rewrote the first paragraph of the introduction to remove the confusion between the aforementioned geophysical and mathematical problems treated in the manuscript.

Page 1, line 27: Minor grammatical fix.

15 Page 2, line 16: Rewrote the paragraphs of the Introduction outlining the aforementioned geophysical and mathematical problems to remove confusion and to create an accurate problem statement.

Page 2, line 27: "flood-fill algorithms" instead of "algorithms".

Page 3, line 2: Moved the two paragraphs justifying the choice of a 1 km resolution to the Results section, where they are now following the paragraph describing the ETOPO1 DEM.

20 Page 3, line 5: Changed the phrasing of the paragraph describing the default algorithm by Zhu et al., improving the clarity of the explanation.

Page 4, line 6: Rewrote the paragraph describing the difference in complexity between filling the sea and filling a lake.

Page 4, line 31: Added reference "as described above" to the term "default algorithm" to clarify which algorithm is meant.

Page 5, line 3: Several minor changes in phrasing to the paragraphs describing the low-resolution block inspection method to 25 improve clarity along the same lines as for the previous section.

Page 5, line 30: Similar changes to the paragraphs describing the shoreline memory improvement.

Page 6, line 9: Similar changes to the paragraphs describing the low resolution lake depth estimation.

Page 7, line 14: Moved the two paragraphs about the 1 km resolution here.

Page 13, line 1: Changed Figure 1, replacing the semi-transparent overlay depicting the true/false-mask by a solid black line 30 depicting the shoreline of said mask. Three additional panels were added. Expanded the caption to give a more comprehensive explanation of the way the algorithm works.

Page 14, line 1: Changed Figure 2 in the same manner.

[revised manuscript text omitted]